# Total Healing of a Partial Rupture of the Supraspinatus Tendon Using Barbotage Technique Associated with Platelet-Rich Plasma: A Randomized, Controlled, and Double-Blind Clinical Trial

**DOI:** 10.3390/biomedicines11071849

**Published:** 2023-06-27

**Authors:** Renato Luiz Bevilacqua de Castro, Breno Pazinatto Antonio, Gustavo Atra Giovannetti, Joyce Maria Annichino-Bizzacchi

**Affiliations:** 1Hematology and Hemotherapy Center, University of Campinas, Cidade Universitária, Rua Carlos Chagas 480, Campinas 13083-878, São Paulo, Brazil; renatojoelho@gmail.com (R.L.B.d.C.); joyce@unicamp.br (J.M.A.-B.); 2Center for Tissue Regeneration Studies, Avenida Barão de Itapura 3378, Taquaral, Campinas 13070-300, São Paulo, Brazil; gugiovannetti@gmail.com

**Keywords:** platelet-rich plasma, PRP, barbotage, healing, supraspinatus tendon, tendon, rotator cuff, dry needling

## Abstract

The prevalence of partial rotator cuff tears (PRCTs) is high in the general population. Our hypothesis is that barbotage, when associated with platelet-rich plasma (PRP), is an effective method for healing these tears. The aim of this study was to compare the effects of barbotage with or without PRP on the healing of partial supraspinatus tendon tears (PSTTs). This study assessed the Western Ontario Rotator Cuff Index score and ultrasound (US) images at 6 weeks and 6 months after treatment. Patients in both groups showed clinical improvement, with no significant difference in scores at 6 weeks. However, at 6 months, the PRP group exhibited significant improvement (*p* = 0.019). Both groups experienced a reduction in ST tear size, but the PRP group demonstrated a significant enhancement at 6 weeks and 6 months. In conclusion, the US-guided barbotage technique, whether associated with PRP or saline solution, proved to be an effective treatment for clinical improvement and reduction in the size of PSTT. Better clinical improvement results were observed with PRP at 6 months. The combination of PRP with barbotage was superior in reducing the size of the ST tear at both 6 weeks and 6 months, resulting in complete healing in 79.3% of the tears.

## 1. Introduction

With technological improvements and advanced diagnostic modalities, there has been much progress as to improving our understanding of the pathophysiology of rotator cuff (RC) rupture [1] and enriching the contributions of Codman [2] and Neer [3,4], who described the evolution of an initial tendinopathy with edema, which can result and then progress to a complete rupture of the tendon. With time, the initial inflammation gives way to fibrosis and partial rupture of the tendon thickness to finally produce the total rupture of the tendon. Cadaveric and natural history studies of the pathology show that the prevalence of RC disease increases with age [5,6,7,8,9]. The prevalence of partial rotator cuff tears (PRCTs) is high. In the general population, PRCT varies between 15% and 32%; and in the dominant shoulder of professional overhead athletes, PRCT can reach 40% [10,11]. Most of these tears seem to occur in the supraspinatus tendon (ST) [12], and a study of 306 cadaveric shoulders [8] showed an incidence of 32% of partial-thickness tears and 19% of full-thickness tears. Yamanaka and Fukuda [13] reported an incidence of ST partial-thickness tears and full-thickness lesions of 13% and 7%, respectively, in a group of 249 cadaveric samples. Rupture progression is correlated with the percentage of thickness of the tendon involved and plastic deformation when the forces capable of leading to total rupture start decreasing [14]. In a follow-up of 13.7 months, Massoud et al. observed 114 patients (118 shoulders) who underwent arthroscopic subacromial decompression for the treatment of symptomatic PRCT, with additional surgery required in 25 patients after a median of 13.7 months (3 to 35), and 10 partial tears had progressed in size. Early diagnosis, mainly in the initial stages, is an important tool in handling this condition, as current treatments for full-thickness tears do not have robust evidence of success [15]. Imaging and histological studies have shown that PRCT healing does not occur spontaneously or with non-anatomical procedures [16]. Open or arthroscopic acromioplasty proved to be neither effective nor preventative regarding the progression of the lesion [17,18]. Therefore, treatments for PRCT are mostly conservative [19]. Subacromial injection of anesthetics or corticosteroids to treat patients with persistent symptoms after rehabilitation therapy and non-steroidal anti-inflammatory drugs (NSAIDs) are often used [20]. Although treatment with NSAIDs and corticosteroid injections are used to relieve inflammation and pain in the shoulder, serious gastrointestinal side effects can occur after the prolonged administration of oral NSAIDs [21]. Despite relief in symptoms, corticosteroid injections did not demonstrate efficacy in healing, and, in addition, they caused joint changes and increased tendon fragility, which may worsen the disease [22]. The pathogenesis of PRCT is still controversial and a subject of research, as an attempt to improve the quality of the tendon in order to avoid a total rupture. Matsen described the PRCT curettage with a 000 curette in the insertion of the great tuberosity of the humerus as a type of treatment in small lesions with good mechanical structure [23]. Thus, new approaches are needed to improve the prognosis of PRCT. Rha [24] suggested that the ultrasound-guided barbotage technique, when associated with platelet-rich plasma (PRP), can heal the PRCT, as demonstrated by clinical improvement in tendinopathy, but without the focus on the healing of the tendon. However, this study evaluated only the clinical improvement in tendinopathy, without the focus on the healing of the tendon. The clinical and imaging improvement of ST tendinopathies with PRP injections has current scientific evidence. Even with level-one evidence [25,26,27] regarding clinical symptoms, the healing rates and the influence of a PRP injection on PRCT, when associated with barbotage, are unknown. The objective of this study was to compare the effects of PRP or saline on barbotage for the treatment of partial-rupture ST. Our hypothesis is that barbotage, when associated with PRP, is an effective method for healing of partial supraspinatus tendon tear (PSTT).

## 2. Materials and Methods

### 2.1. Casuistry

This study, a randomized, prospective, double-blind, controlled clinical trial, was carried out in a single center, at the Hematology and Hemotherapy Center of UNICAMP. Patients were called either by an advertisement aired on a local radio or by spontaneous demand, at the UNICAMP Regenerative Medicine outpatient clinic, to contact the researcher by telephone or in person. The advertisement informed them about the selection of patients with symptomatic PRCT. Patients were informed about the details of the study and the inclusion and exclusion criteria. Accordingly, patients who could participate were referred to the outpatient clinic. No procedure was conducted until these measures were taken, including the signing of the free and informed consent form, approved by the Research Ethics Committee of Unicamp (CAAE: 58323716.1.0000.5404 and UTN: U1111-1286-9993). All patients performed a shoulder US to evaluate the longitudinal measurement of the PRCT, carried out by a single blinded sonographer. Inclusion criteria were patients of both genders, aged over 25 years, with PRCT less than or equal to 1 cm on the longitudinal axis assessed by US. Exclusion criteria were patients who underwent any US-guided barbotage procedure or the use of infiltration with corticosteroids in the affected shoulder during the previous 6 months; the presence of another pathology of the shoulder to be treated (fracture or arthritis of rheumatic origin); cervical spine radiculopathy; pregnancy; neoplasia; severe liver disease or nephropathy; autoimmune, inflammatory, or infectious diseases, whether acute or chronic; hypersensitivity to lidocaine; and use of anti-inflammatory drugs or corticosteroids before 6 weeks of PRP application. The exams used for screening were lipid profile (HDL, LDL, total cholesterol, and triglycerides); creatinine; fasting glucose; uric acid; tumor markers (CA-19.9 and CA-125, α-fetoprotein, and CEA); OTG (Oxalacetic Transaminase); PTG (Pyruvic Transaminase); creatine phosphokinase (CPK); urea; complete blood count; erythrocyte sedimentation (ESR); C-reactive protein; TSH; serology for syphilis; hepatitis A, B, and C; and HIV 1 and HIV 2. Patients who were selected but could not be included were referred to other reference outpatient clinics. The joints of the included patients were randomized into Groups A and B by drawing lots of opaque and sealed envelopes. Group A was those joints submitted to treatment for PRCT using the barbotage technique associated with perilesional and subacromial injection of 6 mL of saline 0,9%. Group B consisted of joints submitted to treatment for PRCT using the barbotage technique associated with perilesional and subacromial injection of 6 mL of PRP.

### 2.2. Guidance on Participation in the Study

The study protocol was explained to all patients as being a study for the treatment of PRCT, using the barbotage technique associated with PRP, or a placebo, and that the product used would be conducted blindly for the applicator, the US executor, and the patient. The use of barbotage and saline as treatment or the placebo in supraspinatus tendinopathy associated with partial tear was described in the previous literature [28,29,30,31,32,33,34,35]. The patients were informed that the evaluations and applications would be carried out at the Regenerative Medicine outpatient clinic of the Hematology and Hemotherapy Center of UNICAMP. Questionnaires, a physical examination, and US were performed at the initial time (T0), at 6 weeks (T6w), and at 6 months (T6m). Patients were provided with information regarding the pathology and post-procedure care, such as resting for 48 h after the application, the use of cryotherapy, and exercises. Throughout the study, patients had the option to contact the outpatient clinic or researchers via telephone to report any symptoms or complaints. Additionally, patients were free to return for a data evaluation related to the pathology or application.

### 2.3. Study Evaluations

#### 2.3.1. Evaluation by US

Ultrasonographic analysis was performed in all patients at T0, T6w, and T6m. The patients were examined sitting down, in internal rotation of the shoulder. The evaluation considered the largest longitudinal measurement of the rupture, and the results were expressed in millimeters [24,36]. A General Electric Logic 5 device (Healthcare, Milwaukee, WI, USA) with a 4–12 MHz linear transducer was used.

#### 2.3.2. Pre- and Post-Treatment Assessment with the Western Ontario Rotator Cuff Index (WORC) Questionnaire

For clinical evaluation, the WORC questionnaire was used. The WORC is a self-reported questionnaire with 21 items in 5 life and health domains (Physical Symptoms, Sports/Recreation, Work, Lifestyle, and Emotions). All items have the same weight, and each has a possible score from 0 to 100 (100 mm Visual Analog Scale). Each domain can be scored separately, and the total score of the questionnaire ranges from 0 to 2100. To make scoring more understandable, the authors of the original version of the WORC recommend that the data be converted to a percentage score by inverting the raw score and converting it to a score out of 100. A score of 0% is the worst score possible, and 100% implies no reduction in health-related quality of life [37]. The questionnaire was applied 03 times, at T0, T6w, and T6m [38]. The Brazilian version of the WORC has proven to be a valid and reliable measurement tool for assessing health-related quality of life in patients with rotator cuff disorders [37].

### 2.4. PRP Preparation and Blinding

The biomedical professional responsible for the study randomly assigned the patients to Group A or B, using the opaque envelope method. Blood was collected from all patients from the peripheral vein in 6 vacuum tubes (BD Vacutainer ACD 8.5 mL) containing ACD anticoagulant, but PRP was prepared only for those assigned to Group B. The blood was centrifuged at 1000× *g* for 10 min and then manipulated inside a sterile vertical laminar flow cabinet, and the upper portion (supernatant) was removed from the top of the plasma layer and discarded. The remainder plasma, approximately 2 mL, termed PRP (liquid format), was collected just above the top of the buffy coat and transferred to a 10 mL syringe, which was used in the treatment. This method produces a recovery of 616,000 platelets per μL and is classified as non-activated PAW-PB β [39,40,41]. After this processing, the PRP was kept at room temperature, without exposure to light. The time between blood collection and application was less than 60 min, and no activation process was used.

A total of 6 mL of 0.9% saline (Group A) or PRP (Group B) was sent to the applicator physician in an opaque syringe.

### 2.5. Technique of the PRP Application Procedure and Follow-Up

Patients were treated in a room dedicated to medical procedures. The injection processes were based on standardized sterile techniques. After the usual sterile preparation with skin asepsis and antisepsis, using chlorhexidine gluconate, skin and subacromial bursa were anesthetized with 2 mL of 1% lidocaine, using a 22G needle, which was inserted into the subacromial space under US guidance. The injured area of the tendon was then perforated 10 times in different planes, and 6 mL of PRP or 0.9% saline solution was injected into the subacromial space, also guided by US (Figure 1). Finally, the area was disinfected, and a blood-stop dressing was applied. The use of an opaque syringe ensured blindness for both the applicator and the patient. The information was available only for a single member of the research group and opened after 6 weeks of the injection. Cryotherapy was recommended to avoid local discomfort. Follow-up was scheduled out to six week and six months after the procedure, in addition to free consultations and telephone contact. After treatment and during the entire follow-up, patients were instructed to report, either by telephone or personally at the Regenerative Medicine outpatient clinic, all side effects or complications related to the treatment.

### 2.6. Rehabilitation

The rehabilitation of patients, regardless of the type of treatment, followed the following procedure:(1)Days 0–2: Cryotherapy for 15 min 4 times a day, rest for activities that use the shoulder, but maintaining routine daily tasks.(2)Days 3–45: Deep heat in the presence of pain, Codman pendular exercises, stretching of the anterior shoulder musculature, and strengthening of the humeral head depressors.(3)Days 45–180: Stretching of the anterior shoulder musculature and strengthening of the humeral head depressors.

### 2.7. Statistical Analysis

#### 2.7.1. Sample Size

Considering the study by Keener et al. in 2015, in which no partial tear of the supraspinatus tendon reached completed healing, and only 1 case out of 54 partially decreased in size in two years, and in view that our pilot study showed that all ruptures decreased in size at 6 months in the PRP group, a single case of decrease would suffice to demonstrate a statistical power of 99% and a significance level of 1% [16]. As the sample calculation presented a small N, we chose to reference our sample calculation by the difference in results between the PRP and placebo groups in a pilot study, which indicated a sample calculation of 8 cases per group (N), with a power of 80% and significance level of 5%.

#### 2.7.2. Analysis Program

Statistical analyzes were evaluated using the GraphPad Prism 9.2 program. Descriptive, comparative, and correlation analyses were used to locate differences between groups. The Wilcoxon test [42] was used for the pre- and post-treatment assessment separately in each group at the studied times. The Mann–Whitney test was used to compare the mean results obtained between the groups, adopting a significance level of *p* < 0.05 for the studied times. To measure the effect size of the treatments, we used the Aw metric, with the following reference values: small (0.56–0.63), medium (0.64–0.70), and large (0.71–1.0). The effect size was calculated using the formula adapted to the Microsoft Excel (Microsoft, Redmond, WA, USA). Effect sizes can be presented associated with statistical significance levels. Since the *p*-values resulting from the results of the statistical tests do not inform magnitude or importance of a difference, the effect sizes (DESs) must be reported [43,44].
Aw = [# (*p* > q) + 0.5# (*p* = q)]/*n*_p_*n*_q_
where # is the counting function, *p* and q are vectors of scores for the two samples, and *n* is the sample size in the group.

## 3. Results

### 3.1. Systematics

During the study selection period, from November 2016 to March 2021, 61 patients were referred for the Regenerative Medicine outpatient clinic: 32 women and 29 men. A total of 24 patients abandoned follow-up before the sixth week: 16 in the placebo group and 8 in the PRP group. The reason for desertion was the COVID-19 pandemic (21 patients), an accident with the treated shoulder fracture (1 patient), lack of transportation (1 patient), and personal reasons (1 patient). From 37 patients that completed the study, 21 were women and 16 men, with a median age of 52 years (29 to 80 years). Of these 37 patients, 8 received injections in both shoulders, 5 with PRP in one side and placebo in the other, 2 with PRP in both shoulders, and 1 with placebo in the 2 shoulders, resulting in a total of 45 treated joints.

### 3.2. Flowchart

All patients answered the WORC questionnaire, but due to the COVID-19 pandemic, during follow-up, some data from the US exam were lost: three US scans not performed at T6w in Group B, two US scans not performed at T6w in Group A, one US scan not performed at T6m in Group B, and one US exam not performed at T6m in Group A. However, this incident did not disturb the calculation by times, according to the significance tests described in the study (Table 1).

### 3.3. Western Ontario Rotator Cuff Index (WORC)

#### 3.3.1. Wilcoxon Test

The results of WORC in Group A showed a significant positive difference in times T0-T6w (*p* = 0.0034) and T0-T6m (*p* = 0.0157), but no difference between the times T6w-T6m (*p* = 0.5336) (Figure 2) was found. In Group B, a significant positive difference in times T0-T6w (*p* < 0.0001) and T0-T6m (*p* < 0.0001) was found, but with no difference between times T6w-T6m (*p* = 0.6974). These data demonstrated a significant improvement in the first 6 weeks after treatment, which was maintained at 6 months (Figure 3).

#### 3.3.2. Mann–Whitney Test

When we compared the two groups (A and B), we found no significant difference in WORC questionnaire at time T0 (*p*-value = 0.7796) or at time T6s (*p*-value = 0.1624), but we found a significant difference at T6m that was favorable to Group B (*p*-value = 0.0199) (Figure 3).

#### 3.3.3. Aw Effect Size

When comparing the results of the WORC questionnaire between the Groups A and B, at T0, we found a negligible effect (Aw = 0.53); at T6w, we found a small effect favorable to Group B (Aw = 0.63); and at T6m, we found a large effect favorable to Group B (Aw = 0.72) (Figure 4).

#### 3.3.4. Significant Clinical Improvement

Wessel et al. standardized the responses to the WORC score applied to RC diseases and suggested 35 points or higher as the level of significant clinical improvement [45]. Considering this level, in the PRP group, 12 out of 30 joints were counted (40%) at T0-T6w, and 14 out of 30 joints (47%) at T0-T6m. In joints that received the placebo, 5 out of 15 joints were counted (33.3%) at T0-T6w, and 2 out of 15 joints (13%) at T0-T6m. We also analyzed the data for the Aw and found a medium effect size of 0.66 favorable to the PRP group at T6w and a large effect size of 0.75 at T6m. The collected data demonstrated that the use of PRP was superior to placebo, with a significant clinical improvement in the WORC score. In Group B, there was a continuous improvement from T6w to T6m, while in Group A, there was an improvement in T6w, which decreased in T6m.

### 3.4. Size of Supraspinatus Tendon Tear

#### 3.4.1. Wilcoxon Test

In Group B, we analyzed the evolution of the improvement in the size of the tendon rupture and found a significant positive difference between T0-T6w (*p*-value < 0.0001), T0 -T6m (*p*-value < 0.0001), and T6w-T6m (*p*-value = 0.0015) (Figure 5). In Group A, we found no significant positive difference at T0-T6w (*p*-value = 0.1953) or T6w-T6m (*p*-value = 0.3652); however, a difference between times T0-T6m (*p*-value = 0.0398) was found (Figure 6).

#### 3.4.2. Size of the Rupture in Groups A and B

In Group B, we demonstrated an overall reduction in tear size in 53.5% of the joints at T0-T6w and in 87.3% of the joints at T0-T6m. In Group A, we found a reduction in tear size in 6.9% of the joints at T0-T6w, and in 32.5% of the joints at T0-T6m.

#### 3.4.3. Mann–Whitney Test

When we compared the two groups, we found no statistical difference at T0 (*p*-value = 0.0754), but the difference was significant at T6s (*p*-value = 0.0119) and at T6m (*p*-value = 0.0002) (Figure 7).

#### 3.4.4. Aw Effect Size

Comparing the results of the improvement in the size of the RC rupture between the two groups, using the Aw tool, a small effect size was demonstrated at T0 (Aw = 0.31), medium at T6w (Aw = 0.68) and large at T6m (Aw = 0.80) (Figure 8).

#### 3.4.5. Complete Healing

In Group B, we found 23 instances of complete healing of the partial supraspinatus tendon tear (PSTT) within 6 months, with a success rate of 79.3%, but in the placebo group, this finding was observed only in 3 joints, with a rate of 21.4% (Figure 9 and Figure 10). No side effects or complications were detected.

## 4. Discussion

Our study is the first randomized, prospective, double-blind, controlled trial to analyze the barbotage technique with or without PRP for the treatment of PSTT, in addition to analyzing the effects of PRP when compared to placebo. We demonstrated that both techniques, barbotage and PRP, or barbotage and saline, resulted in clinical benefits and reduction in the size of the tendon rupture, with superiority to the association with PRP.

### 4.1. Discussion of Results and Comparison with Previous Literature

Few studies published until now have shown scientific evidence regarding the reduction in the size of the PRCT and significant clinical improvement with the treatment of PRP. Kim et al., in 2018, studied 24 patients with PRCT, comparing the clinical evolution of RC rupture with exercise or US-guided intralesional injection of BMAC (bone marrow aspirated concentrated) associated with PRP. Twelve patients received the injection of BMAC-PRP and were advised not to perform physical therapy, and twelve patients in the control group were recommended to perform exercises alone for 3 months. The results showed that BMAC-PRP treatment improved pain and shoulder function in patients with PRCT but did not significantly decrease tear size at 3 weeks or 3 months when compared to the control group. The study has some limitations, such as not being randomized and the injections being performed intralesionally, in addition to the deleterious effects of leukocytes both in the BMAC and the leukocyte-rich PRP that were used [36]. Studies in animal models have shown that intratendinous corticosteroid injections adversely affect the biomechanical properties of tendons [46,47], and the same phenomenon can occur with intralesional PRP injections. Wilson et al. examined the effects of intratendinous injections of PRP, the retention of the injectate within the tendons, the distribution of the intratendinous injectate, and whether intratendinous injection or needle fenestration alter the morphology and mechanical properties of the tendon. The intratendinous injections can alter tendon morphology and mechanics, indicating intralesional injection as a potential cause of tendon rupture, while fenestration alone does not change its mechanical properties [48]. The need for intralesional injection to deliver the therapeutic drug appears to have no support in the current literature. Lundborg et al. demonstrated, in an animal model in 1980, that the primary means of transporting solutions into the tendon is through passive diffusion rather than arterial nutrition [49]. This highlights the importance of the synovial membrane as a producer of nutrients and the ability to deliver soluble substances into the tendon from the peritendinous fluid. Another factor that may have negatively affected the healing results of ST ruptures in Kim’s study was the presence of leukocytes in the injected product. Dragoo et al., in a study with New Zealand white rabbits, compared LR-PRP (PRP rich in leukocytes), LP-PRP (PRP poor in leukocytes), autologous whole blood, and saline injected into patellar tendons. Harvested tendons were stained with hematoxylin and eosin and evaluated semi-quantitatively for total white blood cells (leukocytes). Compared with LP-PRP, LR-PRP caused a significantly higher acute inflammatory response [50]. The method to obtain PRP with a higher number of platelets promotes more contamination with leukocytes, and the inflammatory response with this product does not favor its use. This is corroborated by Boswell et al., who analyzed different PRP formulations regarding the platelet and leukocyte threshold; they showed that reduced leukocyte concentration minimizes catabolic signaling, which is more important than the number of platelets. Furthermore, a maximum biological threshold of benefit has been demonstrated in relation to platelet count, beyond which, further increases in platelet concentration did not result in further anabolic regulation [51]. Cross et al. analyzed LP-PRP and LR-PRP in culture of ST biopsy from 20 patients between the ages of 60 and 80 years old who underwent reverse shoulder arthroplasty. The RNAm expressions of collagen type I (COL1A1), collagen type III (COL3A1), cartilage oligomeric matrix protein (COMP), MMP-9, matrix metalloproteinase-13 (MMP-13), and IL-1b were measured. LP-PRP promoted normal collagen matrix synthesis and decreased cytokines associated with matrix degradation and inflammation to a greater extent than LR-PRP, corresponding to a better healing process [52]. Therefore, in vivo investigations regarding the use of platelets for the treatment of tendinopathy, as well as in vitro characterization of the ideal PRP for tendinopathy and other diseases, are needed.

Rha et al. evaluated 39 patients with tendinosis, including 15 patients with PRCT smaller than 1 cm in the longitudinal axis on an US of the shoulder, in a randomized study, blinded to the patient and open to the applicator. Twenty patients were treated with two PRP injections and US-guided barbotage; however, the intralesional or perilesional target was not identified. Nineteen patients received only US-guided barbotage. The study demonstrated that only the barbotage technique associated with US-guided PRP decreases or heals the PRCT. Unfortunately, the number of patients with PRCT was very small to allow a robust conclusion [24]. PRCT healing with the use of PRP has already been studied by Cai et al. in a randomized, double-blind study that evaluated clinical results at 1 year, in addition to the decrease in the lesion size. The study was performed using MRI [53]. MRI is an expensive and often prohibitive test in Brazil, and its effectiveness compared to US is not a consensus for the assessment of ST rupture. Fotiadou et al. stated that the detection of full-thickness tears of the ST was 98% and 100% for US and MRI, respectively; meanwhile, the accuracy in detecting bursal or joint partial-thickness tears was 87% and 90%, respectively [54]. Jesus et al. stated that there is no statistically significant difference between the sensitivity and specificity of MRI versus US in the diagnosis of total or partial ST ruptures [55]. In addition, Vlychou et al. evaluated the diagnostic performance of US and MRI with surgical findings in symptomatic RCPT and concluded that US imaging can be considered as effective as MRI in detecting RCPT, particularly located in the ST. MRI can be restricted to complex cases in which the delineation of adjacent structures is mandatory before surgical intervention [56]. A point to be considered is that Cai’s study used a 3 Tesla MRI, which detects ST alterations more efficiently. However, Fisher et al., using Cohen’s Kappa and McNemar’s test, evaluated the accuracy of 3 Tesla MRI and US to detect RC and long biceps tendon pathologies confirmed by surgical findings and concluded that both methods are comparable, and the first one should be restricted only to revision cases [57]. The use of MRI in Cai’s study is worthy, but we could not compare it with our findings, as we used US as an imaging parameter. Another characteristic of the study that draws attention is the exclusion criterion for patients over 55 years of age, given that the highest prevalence of the disease is exactly above 60 years of age [58]. Finally, they injected in the subacromial space, without drilling the tendon. Astrom and Rausing in 1995 [59], in a histopathology survey of chronic Achilles tendinopathy, found that a partial tear was always surrounded by a non-inflammatory but degenerated tissue (tendinosis), which would be the generator of the disease, indicating that partial tears are not an independent entity but a complication of tendinosis. Tendon fenestration has been considered a valid treatment for these conditions, and despite the literature supporting the indication, little is known about the associations with other treatments, such as PRP [60]. In 2020, Giovannetti et al. [61], in a systematic review, compared corticosteroid injections to other drugs in the treatment of PRCT, focusing on the effectiveness in terms of pain and functionality of the shoulder, and concluded that none of the techniques undoubtedly prevail. In any case, treatment of PRCT with PRP injections appears to lead to significantly better outcomes in terms of pain and shoulder function at long-term follow-up. It is important to consider that PRP injections were superior only in terms of shoulder function. They found a significant difference only in the Simple Shoulder Test (SST), but not in WORC. However, in the nine studies evaluated, only three analyzed PRP, and all comparing with corticosteroid. In two randomized controlled clinical studies, only one evaluated the WORC score [62], comparing PRP injections, corticosteroid, and prolotherapy. They concluded that, in patients with PRCT, corticosteroid injection provides a short-term relief for pain, function, and quality of life, whereas PRP injection was better in the long term. On the other hand, Kwong et al. [25], in a randomized double-blind study, compared the treatment efficacy of US-guided injections of LP-PRP or corticosteroids and concluded that PRP injections were superior when analyzed by WORC and VAS scores in the short term (3 months) and at 1 year. The healing or decrease in the size of the tears was not evaluated, but only the progression to total tear of the tendon, and with an inexpressive number. Another limitation was that, despite randomization, there were significant differences in baseline outcome scores between the groups. The PRP group started out with more pain and worse patient-reported outcome scores than the corticosteroid group. Despite this, the PRP group achieved better outcomes at 3 months and similar overall outcome scores at 12 months. In 2020, our workgroup published a series of cases, with 26 PRCT smaller than 1 cm, treated with barbotage and US-guided perilesional injection of LP-PRP, and concluded that the treatment can significantly increase the healing rate of the tear and improve shoulder pain and function. Wound healing was measured by US in all 27 tears, resulting in 21 complete healings, 4 downsizing, and 1 unchanged, in addition to a significant clinical improvement on the WORC score. The study did not contemplate the comparison with a control group, and only some of the individuals had US performed by an independent ultra sonographer to verify the reduction of the tear [63]. In the present study, we prospectively evaluated 45 joints with PRCT, with 30 being treated with barbotage associated with PRP and 15 with barbotage and 0.9% saline solution. The PRP formulation used was poor in leukocytes and delivered in the perilesional area, guided by US. We analyzed the improvement in shoulder pain and function measured by the WORC score and decrease of the tear by US measures that showed significant improvement. It is important to point out that the WORC score presents reliability, validity, and responsiveness [64,65,66]. As the WORC score has a language versatility, it is a good tool for international studies and comparisons. In our study, we analyzed the WORC score by using the Wilcoxon test in the first 6 weeks and 6 months after treatment in both groups, but the *p*-value in the first two phases of Group A was higher than the *p*-value calculated for Group B. In addition, we demonstrated the superiority of the results in Group B, which was maintained for 6 months, compared to a positive but less relevant result in Group A, which decreased in 6 months, using the simple percentage of the questionnaire WORC, making it not significant. Wessel et al. standardized the responsiveness of the WORC questionnaire for RC pathologies and suggested that significant clinical improvement in treatments measured by the score needs to be greater than 35 points [45]. In our results, we found that patients who received LP-PRP improved more than 35 points on the WORC score at T0-T6w (12 out of 30 joints; 40%) and at T0-T6m (14 out of 30 joints; 47%). The percentage of patients who received the placebo and improved by more than 35 points in the WORC score was 33.3% at T0-T6w (5 out of 15 joints) and 13% at T0-T6m (2 out of 15 joints). We analyzed the WORC score data by using the Aw effect size and found a mean superiority (Aw effect size = 0.66) that was favorable to the LP-PRP group at T6w and large (Aw effect size = 0.75) at T6m. In Group B, at T6w and T6m, there was a continuous improvement, while in Group A, there was an improvement in T6w, which decreases in T6m. We also evaluated the data and found results of superiority in the Aw effect size in Group B in relation to Group A at T6w and T6m, graded as small and large, respectively. When we compared the results between the LP-PRP and placebo groups, we found that LP-PRP showed great superiority in WORC score rates at 6 months, and it showed smaller ones at 6 weeks. The interpretation of these data showed that, although only barbotage and SF already present a positive result for pain and function, the association of LP-PRP contributes to the longevity of the effect at least for 6 months. Regarding the decrease in size of the PRCT, our results showed that there was a significant reduction of the ST tear in Group B, and this was accentuated at 6 months. On the other hand, although in Group A there was no significant reduction at 6 weeks of treatment, we observed a significant reduction at 6 months. Based on Rha’s study, the barbotage associated with PRP healed a few tears [24], but the sample size was small, and according to Keener’s study about the natural evolution of PSTT, the spontaneous reduction of the lesion is very low or non-existent [16], and the healing of the tears with barbotage and saline in our study is a finding that deserves future scientific evaluations.

When the simple percentage reduction in the size of the tear was used, the results showed that LP-PRP was significantly superior at 6 weeks and 6 months. The effect size of the two techniques was also analyzed by the Aw metric, and the results demonstrated a superiority in the effect size in Group B when compared to the placebo at T6w and T6m, graded in medium and large, respectively. Our study demonstrated that there is a parallelism between the results of improvement in pain and function measured by the WORC score and the measure of the decrease in the size of the ST tear. Although both treatments showed positive results, the association of LP-PRP revealed indisputable superiority. Interestingly, in Group B, out of 29 joints studied, we found 23 complete healings within 6 months, with a success rate for the tear disappearance of 79.3%. Although we observed a positive response in the Group A, the rate was lower, with only 3 total healings out of 14 joints, with a total rupture disappearance rate of 21.4%. These data demonstrate that LP-PRP has the ability to fully heal the PRCT, using the technique of US-guided barbotage. This trend began to occur at 6 weeks in the Group B: in 27 cases, 9 (33.3%) completely healed the tear. Meanwhile, in Group A, at 6 weeks, no tear had healed completely.

Regarding the increase in the size of the tear at T6m, this was not observed in Group B, but 3 (21.4%) out of 14 tears in Group A worsened. The tear deterioration of Group A was expected, since the natural history of PRCT is not healing and even progress to a full-thickness tear.

The LP-PRP demonstrates a proliferative effect that incited natural healing and also acted as a protector against degeneration. Our study is relevant because it presents an effective, safe, and accessible therapy for a prevalent pathology such as PRCT, for which the current treatment shows no robust evidence of success. This may lead to surgical indication or total tear of the ST, with catastrophic consequences for the shoulder, such as chronic pain, decreased function, and arthrosis.

Barbotage and saline induced analgesia as well, explained by the Melzack’s mechanism with hyperstimulation, and the regulation of transcription factors related to proliferation and differentiation caused by fenestration and local bleeding. The results were much better when we associated LP-PRP, a potent analgesic in tendinopathies, with the inducement of a supraphysiological amount of growth factors.

The results were very encouraging and open up a new window of opportunity in the treatment of PRCT, and it is important to highlight that only tendon fenestration was performed, without the deleterious results of intralesional injections. This procedure can only be performed with a US-guided technique and by a physician with proper training.

During the entire follow-up period of the study, no side effects or complications were detected, except for the mild discomfort in the first 48 h after treatment which subsided with the use of cryotherapy for 48 h.

### 4.2. Study Limitations

During the monitoring of patients, due to the COVID-19 pandemic, we had a lack of some US exams: three at T6w in Group B, two at T6w in Group A, one at T6m in Group B, and one at T6m in Group A. However, this incident did not interfere with the calculation using the times, according to the sample size and significance tests described in the study. Another limitation was related to the number of the studied groups, as we only included two groups. The ideal would be to have more groups, without barbotage, with no injection of liquid substances into the shoulder, or both.

Another study limitation was the absence of the MRI method in evaluating PRCT. The use of ultrasound presents an effective alternative to the commonly employed magnetic resonance imaging (MRI). Despite MRI being renowned for its high resolution and detail, the recent literature highlights the potential of ultrasound as an equally effective tool.

Moreover, in a meta-analysis which included 144 diagnostic studies, Liu et al. revealed that high-field MRI had the highest diagnostic value in detecting any type of rotator cuff tendon lesion, followed by low-field MRI, high-frequency ultrasound, low-field MRI, and low-frequency ultrasound. However, the differences in sensitivity and specificity between high-frequency ultrasound and MRI were quite small, and this may be a reason to avoid the potential cost of MRI. Therefore, when making final decisions, the available equipment (MRI and ultrasound parameters) and the examiner’s experience should also be taken into consideration [67].

A salient advantage of ultrasound is its accessibility and expedited operation, which prove to be beneficial in daily situations or in areas where access to MRI equipment is constrained. However, the selection of the optimal diagnostic tool should be primarily guided by the specific clinical scenario at hand.

While MRI maintains its position as the standard reference for assessing PRCT, ultrasound has emerged as a viable and beneficial alternative. More extensive research is required to establish clear guidelines on the most effective usage of ultrasound in the assessment of PRCT healing.

## 5. Conclusions

Our conclusions were that the US-guided barbotage technique associated with LP-PRP or saline is an effective treatment for PRCT, showing clinical improvement and a decrease in the size of the tear. LP-PRP was superior regarding clinical improvement at 6 months, but not at 6 weeks. LP-PRP reduced the size of the PRCT at both 6 weeks and 6 months, reaching complete healing of the tear at 6 months, with a rate of 79.3%. The treatments are safe and have had no complications or side effects. Despite some missing ultrasound exams due to the COVID-19 pandemic, the meaning of the results was unaffected and in accordance with the study’s sample size and significance tests.

## Figures and Tables

**Figure 1 biomedicines-11-01849-f001:**
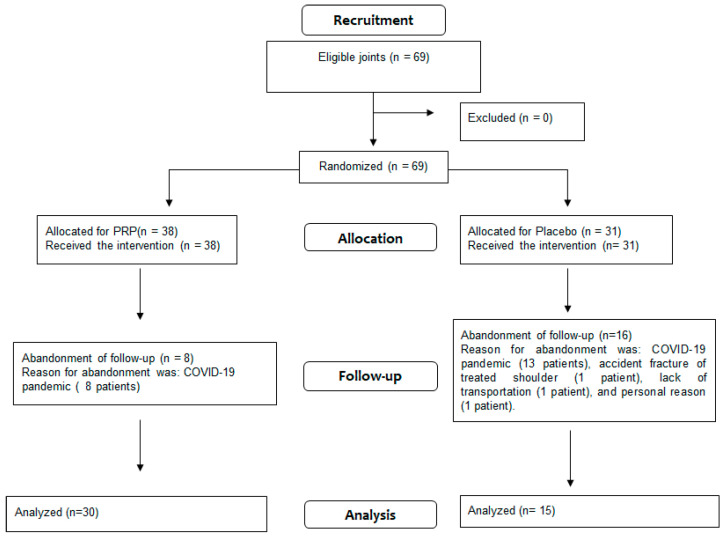
Flowchart containing the recruitment, allocation, follow-up, and analysis.

**Figure 2 biomedicines-11-01849-f002:**
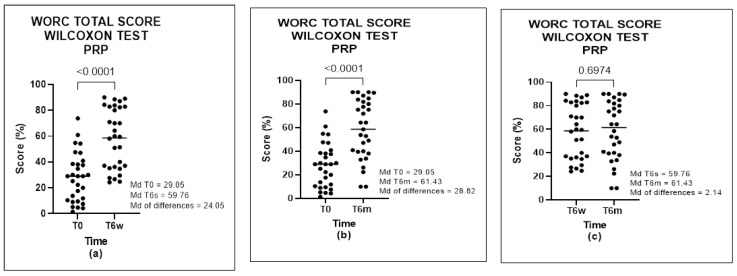
WORC total score of PRP group, analyzed by Wilcoxon Test: (**a**) comparison between T0 and T6w, (**b**) comparison between T0 and T6m, and (**c**) comparison between T6w and T6m.

**Figure 3 biomedicines-11-01849-f003:**
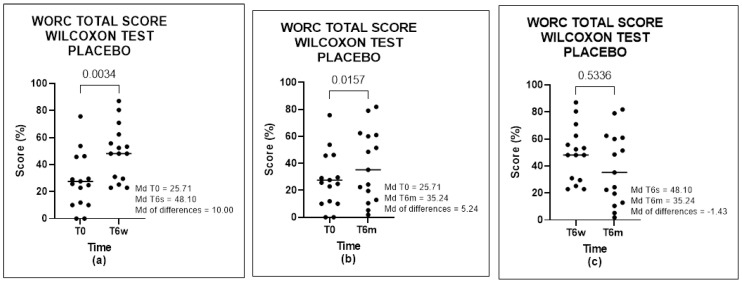
WORC total score of placebo group, analyzed by Wilcoxon Test: (**a**) comparison between T0 and T6w, (**b**) comparison between T0 and T6m, and (**c**) comparison between T6w and T6m.

**Figure 4 biomedicines-11-01849-f004:**
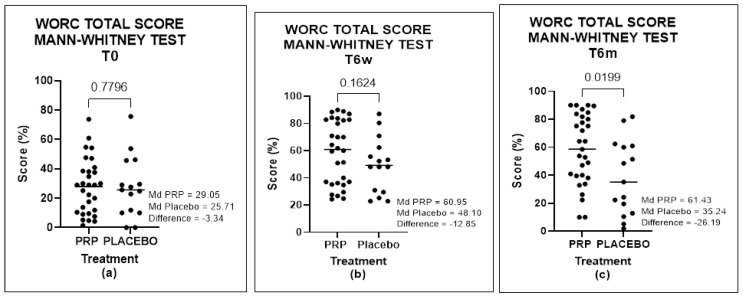
WORC total score of groups at times T0, T6w, and T6m, analyzed by the Mann–Whitney test: (**a**) comparison between Groups A and B at T0, (**b**) comparison between Groups A and B at T6w, and (**c**) comparison between Groups A and B at T6m.

**Figure 5 biomedicines-11-01849-f005:**
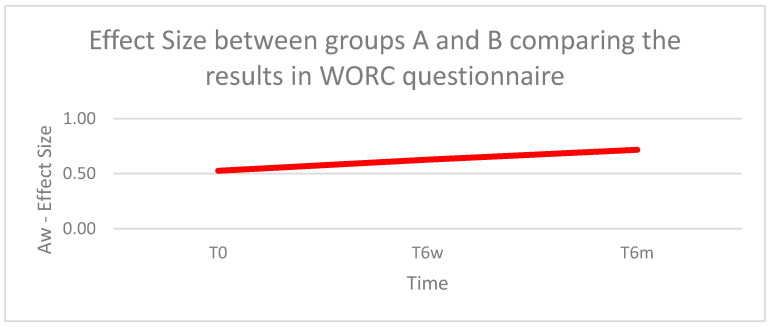
Aw effect size between Groups A and B at times T0, T6w, and T6m, comparing the results of the WORC questionnaire.

**Figure 6 biomedicines-11-01849-f006:**
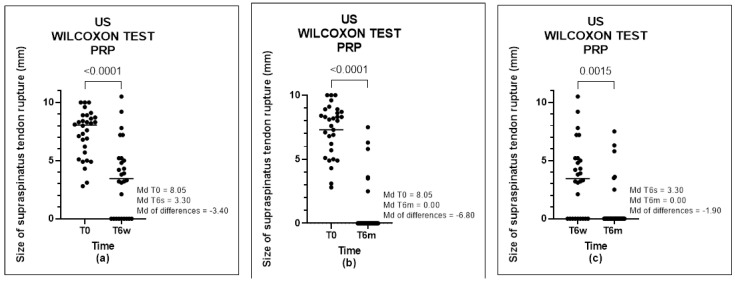
US of PRP group, analyzed by Wilcoxon test: (**a**) comparison between T0 and T6w, (**b**) comparison between T0 and T6m, and (**c**) comparison between T6w and T6m.

**Figure 7 biomedicines-11-01849-f007:**
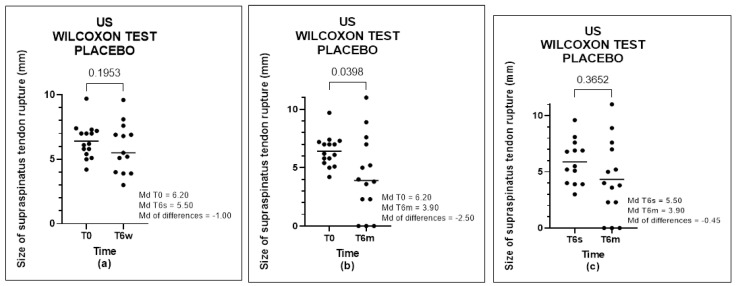
US of placebo group, analyzed by Wilcoxon Test: (**a**) comparison between T0 and T6w, (**b**) comparison between T0 and T6m, and (**c**) comparison between T6w and T6m.

**Figure 8 biomedicines-11-01849-f008:**
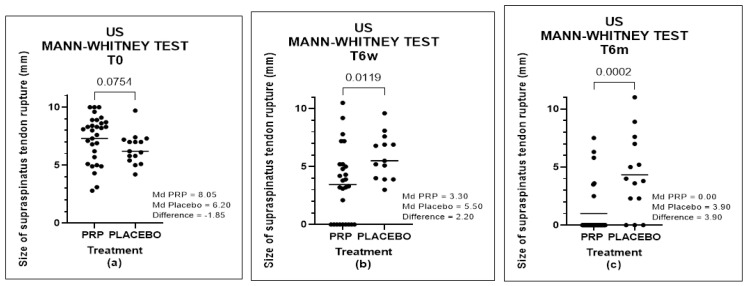
US of groups at times T0, T6w, and T6m, analyzed by Mann–Whitney test: (**a**) comparison between Groups A and B at T0, (**b**) comparison between Groups A and B at T6w, and (**c**) comparison between Groups A and B at T6m.

**Figure 9 biomedicines-11-01849-f009:**
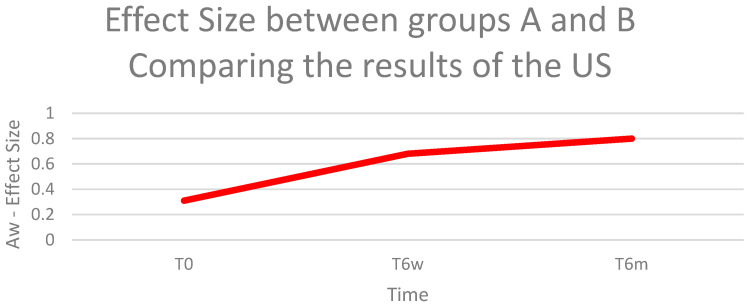
Aw effect size between Groups A and B at times T0, T6w, and T6m, comparing the results of the US.

**Figure 10 biomedicines-11-01849-f010:**
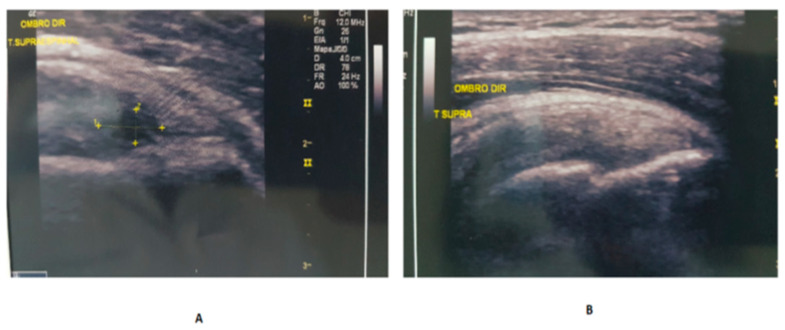
(**A**) PSTT before treatment. (**B**) PSTT total healing after T6m.

**Table 1 biomedicines-11-01849-t001:** Table containing treatment performed on each joint, with size of lesions in millimeters (obtained by ultrasound) at times determined in the study. Cells containing the letter X represent the time the test was not performed.

Joint	Treatment	US T0	US T6w	US T6m
1	PRP	10.0	10.5	0.0
2	PRP	5.1	0.0	3.6
3	PRP	6.8	4.3	0.0
4	PRP	8.0	5.2	3.5
5	PRP	5.7	0.0	0.0
6	PRP	3.1	0.0	0.0
7	PRP	10.0	0.0	0.0
8	PRP	8.6	4.2	X
9	PRP	6.2	0.0	0.0
10	PRP	8.2	0.0	0.0
11	PRP	8.7	5.2	0.0
12	PRP	7.6	7.2	0.0
13	PRP	8.3	3.1	2.5
14	PRP	7.3	0.0	0.0
15	PRP	6.9	3.2	0.0
16	PRP	7.1	0.0	0.0
17	PRP	8.4	5.0	5.8
18	PRP	8.9	X	0.0
19	PRP	8.1	4.8	7.5
20	PRP	10.0	9.2	6.3
21	PRP	9.1	7.2	0.0
22	PRP	4.9	3.2	0.0
23	PRP	5.0	0.0	0.0
24	PRP	4.9	3.8	0.0
25	PRP	8.9	X	0.0
26	PRP	8.3	7.8	0.0
27	PRP	2.8	3.3	0.0
28	PRP	4.3	2.1	0.0
29	PRP	9.6	3.9	0.0
30	PRP	8.3	X	0.0
31	Placebo	7.0	3.9	4.0
32	Placebo	5.0	4.0	3.8
33	Placebo	7.2	8.1	5.2
34	Placebo	7.0	5.5	7.0
35	Placebo	5.4	9.6	0.0
36	Placebo	6.2	X	0.0
37	Placebo	4.2	5.1	2.3
38	Placebo	7.3	5.2	3.6
39	Placebo	7.4	6.8	8.9
40	Placebo	7.0	7.6	7.6
41	Placebo	5.8	3.0	2.3
42	Placebo	5.8	X	0.0
43	Placebo	9.7	6.9	5.0
44	Placebo	5.1	3.9	11.0
45	Placebo	6.1	6.9	X

## Data Availability

Data are available with the corresponding authors upon reasonable request.

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
