# Peer review of "Total Healing of a Partial Rupture of the Supraspinatus Tendon Using Barbotage Technique Associated with Platelet-Rich Plasma: A Randomized, Controlled, and Double-Blind Clinical Trial"

_biomedicines, 2023, doi:10.3390/biomedicines11071849_

Round 1
Reviewer 1 Report
The following are major conceptual and technical issues regarding this paper:
Abstract
· The problem that is treated and the hypothesis should be defined
Introduction
· “Technology has improved…” – explain what technology and how it has improved?
· “Imaging and histological 49 studies have shown that PRCT healing does not occur spontaneously, or with non- anatomical procedures.” – provide reference
· “Rha 64 [22] demonstrated that the ultrasound-guided barbotage technique, when associated with platelet rich plasma (PRP), reduced, or healed the PRCT. However, this study only evaluated the clinical improvement in tendinopathy, without the focus on the healing of the tendon.” - these two sentences have a mutual contradiction. Healed or not healed? – Needs explanation
· There is no exact hypothesis.
Methods
· Barbotage technique is indicated for the treatment of calcific tendinitis. Why should it be used for the treatment of RC tears? If you have any report for the previous RC tears treatment by barbotage by Saline alone, please give a reference, and if not, it will be unethical to have this control group.
· 40% of the study group dropped from the study – this point should be discussed in correlation to the study outcome and conclusions because it raises an important uncertainty about the meaning of the results.
· US accuracy in evaluating supraspinatus tears is about 84%, therefore at least 15% of the tears might be misdiagnosed by US [JRPMS | September 2022 | Vol. 6, No. 3 | 72-82]. How this fact affects the outcome of this study?
· US has low sensitivity for detecting supraspinatus tears [https://doi.org/10.1590/1516-3180.2018.0069170418 ] how this affects the conclusions on the reliability of the results of this study? Was this an appropriate diagnostic modality for evaluating the size and healing of the tendon?
· “WORC questionnaire” – description and reliability of the score should be given
Results
· Table 3 – in 3 of 15 placebo treatments (20%) – injection by Saline caused complete healing of RC tear. Isn’t it strange? Does the spontaneous healing of partial RC tear possible? Is this supported by the known pathophysiology on the healing of RC tear, or rather raises the question of the reliability of the US evaluation?
The English style is appropriate
Author Response
Response to Reviewer 1 Comments
Point 1: The problem that is treated and the hypothesis should be defined (Abstract).
Response 1: The problem that is treated and the hypothesis were defined and added to the abstract.
“The prevalence of partial rotator cuff tear (PRCT) is high in the general population. Our hypothesis is that barbotage, when associated with Platelet Rich Plasma (PRP), is an effective method for healing these tears”.
Point 2: “Technology has improved…” (Introduction) – explain what technology and how it has improved?
Response 2: The technology was explained and a new reference was added to the study.
“With technological improvements and advanced diagnostic modalities, there has been much progress as to improved understanding of the pathophysiology of rotator cuff (RC) rupture [1]”.
Reference:
- Eckers F.; Loske S.; Ek E.T.; Müller A.M. Current understanding and new advances in the surgical management of reparable rotator cuff tears: A scoping review. J Clin Med 2023, 12(5), 1713.
Point 3: “Imaging and histological 49 studies have shown that PRCT healing does not occur spontaneously, or with non- anatomical procedures.” (Introduction) – provide reference.
Response 3: The reference was provided.
Reference:
- Keener J.D.; Hsu J.E.; Steger-May K.; Teefey S.A.; Chamberlain A.M.; Yamaguchi K. Patterns of tear progression for asymp-tomatic degenerative rotator cuff tears. J Shoulder Elbow Surg 2015, 24(12), 1845-1851.
Point 4: “Rha [24] demonstrated that the ultrasound-guided barbotage technique, when associated with platelet rich plasma (PRP), reduced, or healed the PRCT. However, this study only evaluated the clinical improvement in tendinopathy, without the focus on the healing of the tendon.” (Introduction) - these two sentences have a mutual contradiction. Healed or not healed? – Needs explanation
Response 4: The sentence has been redrafted to clarify and end the contradiction.
“Rha suggested that the ultrasound-guided barbotage technique, when associated with platelet rich plasma (PRP), can heal the PRCT, demonstrated by clinical improvement in tendinopathy, but without the focus on the healing of the tendon”.
Reference:
- Rha D.W.; Park G.Y.; Kim Y.K.; Kim M.T.; Lee S.C. Comparison of the therapeutic effects of ultrasound-guided platelet-rich plasma injection and dry needling in rotator cuff disease: a randomized controlled trial. Clin Rehabil 2013, 27(2), 113-122.
Point 5: There is no exact hypothesis (Introduction).
Response 5: The hypothesis was added to the introduction.
“Our hypothesis is that barbotage, when associated with PRP, is an effective method for healing of partial supraspinatus tendon tear (PSTT)”.
Point 6: “Barbotage technique is indicated for the treatment of calcific tendinitis” (Methods). Why should it be used for the treatment of RC tears? If you have any report for the previous RC tears treatment by barbotage by Saline alone, please give a reference, and if not, it will be unethical to have this control group.
Response 6: The reference was provided.
“The use of barbotage and saline as treatment or placebo in supraspinatus tendinopathy associated with partial tear has been described in the previous literature [28–36]”.
Reference:
- Moghtaderi A.; Sajadiyeh S.; Khosrawi S.; Dehghan F.; Bateni V. Effect of subacromial sodium hyaluronate injection on rotator cuff disease: A double-blind placebo-controlled clinical trial. Adv Biomed Res 2013, 30(2), 89.
- Coombes B.K.; Bisset L.; Vicenzino B. Efficacy and safety of corticosteroid injections and other injections for management of tendinopathy: a systematic review of randomised controlled trials. Lancet 2010, 20, 376(9754), 1751-1767.
- Kleinhenz J.; Streitberger K.; Windeler J.; Güssbacher A.; Mavridis G.; Martin E. Randomised clinical trial comparing the effects of acupuncture and a newly designed placebo needle in rotator cuff tendinitis. Pain 1999, 83(2), 235-241.
- Kesikburun S.; Tan A.K.; Yilmaz B.; YaÅŸar E.; YazicioÄŸlu K. Platelet-rich plasma injections in the treatment of chronic rotator cuff tendinopathy: a randomized controlled trial with 1-year follow-up. Am J Sports Med 2013, 41(11), 2609-2616
- Rha D.W.; Park G.Y.; Kim Y.K.; Kim M.T.; Lee S.C. Comparison of the therapeutic effects of ultrasound-guided platelet-rich plasma injection and dry needling in rotator cuff disease: a randomized controlled trial. Clin Rehabil 2013, 27(2), 113-122.
- Penning L.I.; de Bie R.A.; Walenkamp G.H. Subacromial triamcinolone acetonide, hyaluronic acid and saline injections for shoulder pain an RCT investigating the effectiveness in the first days. BMC Musculoskelet Disord 2014, 23(15), 352.
- Akpinar S.; Hersekli M.A.; Demirors H.; Tandogan R.N.; Kayaselcuk F. Effects of methylprednisolone and betamethasone injections on the rotator cuff: an experimental study in rats. Adv Ther 2002, 19(4), 194-201.
- Housner J.A.; Jacobson J.A.; Misko R. Sonographically guided percutaneous needle tenotomy for the treatment of chronic tendinosis. J Ultrasound Med 2009, 28(9), 1187-1192.
- Finnoff J.T.; Fowler S.P.; Lai J.K.; Santrach P.J.; Willis E.A.; Sayeed Y.A.; Smith J. Treatment of chronic tendinopathy with ultrasound-guided needle tenotomy and platelet-rich plasma injection. PM R 2011, 3(10), 900-911.
Point 7: 40% of the study group dropped from the study (Methods) – this point should be discussed in correlation to the study outcome and conclusions because it raises an important uncertainty about the meaning of the results.
Response 7: This point was added to the study limitations and conclusion.
Study limitations: “During the monitoration of patients, due to the COVID-19 pandemic, we had a lack of some US exams: 3 at T6w in Group B, 2 at T6w in Group A, 1 at T6m in Group B, and 1 at T6m in Group A. However, this incident did not interfere with the calculation by times, according to the sample size and significance tests described in the study. Another limitation was related to the number of the studied groups, as we only included two groups. The ideal would be to have more groups, without barbotage, or with no injection of liquid substances into the shoulder, or both”.
Conclusion: “Despite some missing ultrasound exams due to the COVID-19 pandemic, the meaning of the results was unaffected and in accordance with the study's sample size and significance tests”.
Point 8: US accuracy in evaluating supraspinatus tears is about 84%, therefore at least 15% of the tears might be misdiagnosed by US [JRPMS | September 2022 | Vol. 6, No. 3 | 72-82] (Methods). How this fact affects the outcome of this study?
Response 8: Based on the information provided in the article "Sensitivity and specificity of ultrasonography in diagnosing supraspinatus lesions: a prospective accuracy diagnostic study" by Yazigi Junior et al. [1], the authors aimed to evaluate the accuracy of shoulder ultrasound in diagnosing supraspinatus tendon tears, using magnetic resonance imaging (MRI) as the reference standard. The study included 80 patients with shoulder pain who underwent both ultrasound and MRI.
The results of the study showed that ultrasound had low sensitivity (36.3%) but high specificity (91.7%) in detecting supraspinatus tears overall. For partial tears, the sensitivity was 25.8%, and for complete tears, the sensitivity was 46.2% with a specificity of 100%. The study also found high accuracy (91.3%) and specificity (100%) in diagnosing complete tears [1].
It is important to note that the study had some limitations, as mentioned by the authors themselves. The ultrasounds were performed by radiologists who were not specialists in musculoskeletal disorders, which may have influenced the results. Additionally, the study had a small sample size and was conducted at a single center, which may limit the generalizability of the results. Out of the 80 evaluated patients, 6 (7.5%) had no tendon lesions, 30 (37.5%) had tendinopathy, 31 (38.75%) had partial tears, and 13 (16.25%) had complete tears of the supraspinatus tendon [1]. Furthermore, the study was conducted with 3 Tesla Magnetic Resonance Imaging and did not use shoulder arthroscopy as a comparison method.
To fully assess the validity of these results and analyze the overall consensus in the literature, it would be necessary to review a broader range of studies and systematic reviews on the sensitivity and specificity of ultrasound in diagnosing supraspinatus tendon lesions.
On the other hand, studies such as Fotiadou et al. [2], Jesus et al. [3], and Vlychou et al. [4] state that comparing the results of ultrasound and MRI with arthroscopy is considered the gold standard for evaluating the sensitivity and specificity of these methods in diagnosing rotator cuff tendon lesions.
These studies indicate that there is no statistically significant difference between the sensitivities and specificities of MRI compared to ultrasound in diagnosing partial or complete supraspinatus tendon tears [2-4]. Recent studies, such as Xin Ooi et al. [5] and Gyftopoulos et al. [6], have shown that ultrasound and MRI have similar sensitivity and specificity in evaluating the rotator cuff.
In a meta-analysis, Liu et al., which included 144 diagnostic studies, revealed that high-field MRI had the highest diagnostic value in detecting any type of rotator cuff tendon lesion, followed by low-field MRI, high-frequency ultrasound, low-field MRI, and low-frequency ultrasound. However, the differences in sensitivity and specificity between high-frequency ultrasound and MRI were quite small, which may be a reason to avoid the potential cost of MRI. Therefore, when making final decisions, the available equipment (MRI and ultrasound parameters) and the examiner's experience should also be taken into consideration [7].
Regarding postoperative evaluation of rotator cuff repair surgery, studies have shown that both MRI and ultrasound are equally valid imaging options for detecting possible re-tears of the rotator cuff after previous repair. A systematic review and meta-analysis by Gyftopoulos et al. found no significant difference between ultrasound and MRI in diagnosing rotator cuff tendon tears after previous repair. Therefore, both MRI and ultrasound can be considered first-line imaging options for evaluating suspected re-tears of the rotator cuff after surgical repair [6].
Based on this evidence, it can be stated that ultrasound is not inferior to MRI in terms of sensitivity in the diagnosis and monitoring of partial supraspinatus tendon tears. Both imaging modalities can play an important role in detecting and monitoring these injuries, with arthroscopy being considered the gold standard for result comparison [2-4].
In summary, although ultrasound (US) may have relatively lower sensitivity compared to magnetic resonance imaging (MRI), studies demonstrate that it has high specificity and can be a valid alternative for diagnosing and monitoring partial supraspinatus tendon tears. Comparing the results of US and MRI with arthroscopy is considered the gold standard for assessing the sensitivity and specificity of these methods [2-4].
It is important to note that the choice between US and MRI may depend on various factors, including equipment availability, examiner experience, and the need for a more detailed assessment of adjacent structures. US may be a more accessible and convenient option in terms of availability and cost, especially in contexts where MRI may be limited [1-4].
However, it is crucial that US is performed by experienced professionals, as was the case in our study, where the sonographer was blinded to the treatments and specialized in musculoskeletal ultrasound.
References not cited in the study, only for the reviewer:
- Yazigi Junior J, Reis FBL, D'Ippolito G. Sensitivity and specificity of ultrasonography in diagnosing supraspinatus lesions: a prospective accuracy diagnostic study. Rev Bras Ortop. 2018;53(4):443-448.
- Fotiadou AN, Vlychou M, Papadopoulos P, et al. Ultrasonography of symptomatic rotator cuff tears compared with MR imaging and surgery. Eur J Radiol. 2008;68(1):174-179.
- Jesus JO, Parker L, Frangos AJ, Nazarian LN. Accuracy of MRI, MR arthrography, and ultrasound in the diagnosis of rotator cuff tears: a meta-analysis. AJR Am J Roentgenol. 2009;192(6):1701-1707.
- Vlychou M, Papadopoulos P, Zampeli F, et al. Symptomatic partial rotator cuff tears: diagnostic performance of ultrasound and magnetic resonance imaging with surgical correlation. Acta Radiol. 2009;50(1):101-105.
- Xin Ooi MW, Fenning L, Dhir V, Basu S. Rotator cuff assessment on imaging. J Clin Orthop Trauma. 2021;18:121-135.
- Gyftopoulos S, Cardoso MDS, Rodrigues TC, Qian K, Chang CY. Postoperative Imaging of the Rotator Cuff: A Systematic Review and Meta-Analysis. AJR Am J Roentgenol. 2022;219(5):717-723.
- Liu F, Dong J, Shen WJ, Kang Q, Zhou D, Xiong F. Detecting Rotator Cuff Tears: A Network Meta-analysis of 144 Diagnostic Studies. Orthop J Sports Med. 2020 5;8(2)
Point 9: US has low sensitivity for detecting supraspinatus tears [https://doi.org/10.1590/1516-3180.2018.0069170418 ] (Methods). How this affects the conclusions on the reliability of the results of this study? Was this an appropriate diagnostic modality for evaluating the size and healing of the tendon?
Response 9: Based on the information provided in the article "Sensitivity and specificity of ultrasonography in diagnosing supraspinatus lesions: a prospective accuracy diagnostic study" by Yazigi Junior et al. [1], the authors aimed to evaluate the accuracy of shoulder ultrasound in diagnosing supraspinatus tendon tears, using magnetic resonance imaging (MRI) as the reference standard. The study included 80 patients with shoulder pain who underwent both ultrasound and MRI.
The results of the study showed that ultrasound had low sensitivity (36.3%) but high specificity (91.7%) in detecting supraspinatus tears overall. For partial tears, the sensitivity was 25.8%, and for complete tears, the sensitivity was 46.2% with a specificity of 100%. The study also found high accuracy (91.3%) and specificity (100%) in diagnosing complete tears [1].
It is important to note that the study had some limitations, as mentioned by the authors themselves. The ultrasounds were performed by radiologists who were not specialists in musculoskeletal disorders, which may have influenced the results. Additionally, the study had a small sample size and was conducted at a single center, which may limit the generalizability of the results. Out of the 80 evaluated patients, 6 (7.5%) had no tendon lesions, 30 (37.5%) had tendinopathy, 31 (38.75%) had partial tears, and 13 (16.25%) had complete tears of the supraspinatus tendon [1]. Furthermore, the study was conducted with 3 Tesla Magnetic Resonance Imaging and did not use shoulder arthroscopy as a comparison method.
To fully assess the validity of these results and analyze the overall consensus in the literature, it would be necessary to review a broader range of studies and systematic reviews on the sensitivity and specificity of ultrasound in diagnosing supraspinatus tendon lesions.
On the other hand, studies such as Fotiadou et al. [2], Jesus et al. [3], and Vlychou et al. [4] state that comparing the results of ultrasound and MRI with arthroscopy is considered the gold standard for evaluating the sensitivity and specificity of these methods in diagnosing rotator cuff tendon lesions.
These studies indicate that there is no statistically significant difference between the sensitivities and specificities of MRI compared to ultrasound in diagnosing partial or complete supraspinatus tendon tears [2-4]. Recent studies, such as Xin Ooi et al. [5] and Gyftopoulos et al. [6], have shown that ultrasound and MRI have similar sensitivity and specificity in evaluating the rotator cuff.
In a meta-analysis, Liu et al., which included 144 diagnostic studies, revealed that high-field MRI had the highest diagnostic value in detecting any type of rotator cuff tendon lesion, followed by low-field MRI, high-frequency ultrasound, low-field MRI, and low-frequency ultrasound. However, the differences in sensitivity and specificity between high-frequency ultrasound and MRI were quite small, which may be a reason to avoid the potential cost of MRI. Therefore, when making final decisions, the available equipment (MRI and ultrasound parameters) and the examiner's experience should also be taken into consideration [7].
Regarding postoperative evaluation of rotator cuff repair surgery, studies have shown that both MRI and ultrasound are equally valid imaging options for detecting possible re-tears of the rotator cuff after previous repair. A systematic review and meta-analysis by Gyftopoulos et al. found no significant difference between ultrasound and MRI in diagnosing rotator cuff tendon tears after previous repair. Therefore, both MRI and ultrasound can be considered first-line imaging options for evaluating suspected re-tears of the rotator cuff after surgical repair [6].
Therefore, based on this evidence, it can be stated that ultrasound is not inferior to MRI in terms of sensitivity in the diagnosis and monitoring of partial supraspinatus tendon tears. Both imaging modalities can play an important role in detecting and monitoring these injuries, with arthroscopy being considered the gold standard for result comparison [2-4].
In summary, although ultrasound (US) may have relatively lower sensitivity compared to magnetic resonance imaging (MRI), studies demonstrate that it has high specificity and can be a valid alternative for diagnosing and monitoring partial supraspinatus tendon tears. Comparing the results of US and MRI with arthroscopy is considered the gold standard for assessing the sensitivity and specificity of these methods [2-4].
It is important to note that the choice between US and MRI may depend on various factors, including equipment availability, examiner experience, and the need for a more detailed assessment of adjacent structures. US may be a more accessible and convenient option in terms of availability and cost, especially in contexts where MRI may be limited [1-4].
However, it is crucial that US is performed by experienced professionals, as was the case in our study, where the sonographer was blinded to the treatments and specialized in musculoskeletal ultrasound.
References not cited in the study, only for the reviewer:
- Yazigi Junior J, Reis FBL, D'Ippolito G. Sensitivity and specificity of ultrasonography in diagnosing supraspinatus lesions: a prospective accuracy diagnostic study. Rev Bras Ortop. 2018;53(4):443-448.
- Fotiadou AN, Vlychou M, Papadopoulos P, et al. Ultrasonography of symptomatic rotator cuff tears compared with MR imaging and surgery. Eur J Radiol. 2008;68(1):174-179.
- Jesus JO, Parker L, Frangos AJ, Nazarian LN. Accuracy of MRI, MR arthrography, and ultrasound in the diagnosis of rotator cuff tears: a meta-analysis. AJR Am J Roentgenol. 2009;192(6):1701-1707.
- Vlychou M, Papadopoulos P, Zampeli F, et al. Symptomatic partial rotator cuff tears: diagnostic performance of ultrasound and magnetic resonance imaging with surgical correlation. Acta Radiol. 2009;50(1):101-105.
- Xin Ooi MW, Fenning L, Dhir V, Basu S. Rotator cuff assessment on imaging. J Clin Orthop Trauma. 2021;18:121-135.
- Gyftopoulos S, Cardoso MDS, Rodrigues TC, Qian K, Chang CY. Postoperative Imaging of the Rotator Cuff: A Systematic Review and Meta-Analysis. AJR Am J Roentgenol. 2022;219(5):717-723.
- Liu F, Dong J, Shen WJ, Kang Q, Zhou D, Xiong F. Detecting Rotator Cuff Tears: A Network Meta-analysis of 144 Diagnostic Studies. Orthop J Sports Med. 2020 Feb 5;8(2)
Point 10: “WORC questionnaire” (Methods) – description and reliability of the score should be given.
Response 10: The description and reliability of the score was added to the study.
“The WORC is a self-reported questionnaire with 21 items in 5 life and health domains (Physical Symptoms, Sports/Recreation, Work, Lifestyle, Emotions). All items have the same weight, and each has a possible score from 0 to 100 (100 mm Visual Analog Scale). Each domain can be scored separately, and the total score of the questionnaire ranges from 0 to 2100. To make scoring more understandable, the authors of the original version of the WORC recommend that the data be converted to a percentage score by inverting the raw score and converting it to a score out of 100. A score of 0% is the worst score possible, and 100% implies no reduction in health–related quality of life [38]. The Brazilian version of the WORC has proven to be a valid and reliable measurement tool for assessing health-related quality of life in patients with rotator cuff disorders [38]”.
Reference:
- Lopes A.D.; Ciconelli R.M.; Carrera E.F.; Griffin S.; Faloppa F.; Dos Reis F.B. Validity and reliability of the Western Ontario Rotator Cuff Index (WORC) for use in Brazil. Clin J Sport Med 2008, 18(3), 266-272.
Point 11: Table 3 – in 3 of 15 placebo treatments (20%) – injection by Saline caused complete healing of RC tear (Results). Isn’t it strange? Does the spontaneous healing of partial RC tear possible? Is this supported by the known pathophysiology on the healing of RC tear, or rather raises the question of the reliability of the US evaluation?
Response 11: The finding of complete healing with the association of saline and barbotage was cited with greater emphasis in the study, being a finding that deserves future scientific evaluations.
“Based on Rha's study, the barbotage associated with PRP healed a few tears [24], but the sample size was small, and according to Keener’s study about the natural evolution of PSTT, the spontaneous reduction of the lesion is very low or non-existent [16], and the healing of the tears with barbotage and saline in our study is a finding that deserves future scientific evaluations”.
Reference:
- Keener J.D.; Hsu J.E.; Steger-May K.; Teefey S.A.; Chamberlain A.M.; Yamaguchi K. Patterns of tear progression for asymp-tomatic degenerative rotator cuff tears. J Shoulder Elbow Surg 2015, 24(12), 1845-1851
- Rha D.W.; Park G.Y.; Kim Y.K.; Kim M.T.; Lee S.C. Comparison of the therapeutic effects of ultrasound-guided platelet-rich plasma injection and dry needling in rotator cuff disease: a randomized controlled trial. Clin Rehabil 2013, 27(2), 113-122.
Reviewer 2 Report
Interesting and well designed study. Methods are well explained, supporting duration of the study. Introduction explained all key messages. Ultrasound machine is pretty old, since the beginning of the study was in 2016. Did authors use other machine later? Do you possibly have a shear wave elastography results? Those would be interesting to check in such ruptures and healing process monitoring.
Discussion section should be significantly shortened. Reference list should contain more recent literature.
Interesting and well designed study. Methods are well explained, supporting duration of the study. Introduction explained all key messages. Ultrasound machine is pretty old, since the beginning of the study was in 2016. Did authors use other machine later? Do you possibly have a shear wave elastography results? Those would be interesting to check in such ruptures and healing process monitoring.
Discussion section should be significantly shortened. Reference list should contain more recent literature.
Author Response
Response to Reviewer 2 Comments
Point 1: Ultrasound machine is pretty old, since the beginning of the study was in 2016. Did authors use other machine later?
Response 1: No other machine was used in the study, and in the time of the exam that machine was usual in brazilian scientific community.
Point 2: Do you possibly have a shear wave elastography results?
Response 2: Unfortunately no, because we did not support this technology at the time of the study. We will develop a new study with compression elastography in healed supraspinatus tendon.
Point 3: Discussion section should be significantly shortened.
Response 3: The discussion section was shortened by removing the following sentences:
“In the first part of the study, eighteen lamb extensor tendons were selected to receive injec-tion of PRP containing methylene blue (PRP /MB), injection of only methylene blue (MB), or needle fenestration. The volume of retained injectate was measured and the distribution of injectate and tendon morphology were examined microscopically. In the second part of the study, eighteen porcine flexor tendons were divided into control, needle fenestration or saline injection groups. Young's Modulus was then determined for each tendon under 0-4% tension. The intratendinous injection was retained within the tendon, and the dif-ference between the PRP and PRP/MB groups was not significant (p = 0.78). Intratendi-nous spread of the injectable solution within the tendon occurred mainly in a proximal to distal direction, with very little transverse penetration. Intratendinous injections resulted in disruption of microscopic morphology (separation and disorganization of collagen bundles and cellular distribution). There were significant differences in Young's Modulus between the control (Ectrl = 2415.48 Kpa) and the injected tendons (Einj = 1753.45 Kpa) with 4% strain (p = 0.01). There were no differences in Young's Modulus between the fe-nestrated and control tendons, demonstrating that intratendinous injections of PRP are retained within the tendon and are mainly distributed longitudinally with minimal transverse spread”.
“Mononuclear cells (macrophages and lymphocytes), polymorphonuclear cells (PMNs), vascularity, fiber structure, and fibrosis.
“Transforming growth factor b – 1 (TGFb-1), interleukin-1b (IL-1b), interleukin-1 receptor antagonist (IL-1Ra), interleukin-6 (IL-6), interleukin-8 (IL-8) and matrix metalloprotein-ase-9 (MMP-9) were quantified”.
Point 4: Reference list should contain more recent literature.
Response 4: More recent literature were added to the study.
- Eckers F.; Loske S.; Ek E.T.; Müller A.M. Current understanding and new advances in the surgical management of reparable rotator cuff tears: A scoping review. J Clin Med 2023, 12(5), 1713.
Round 2
Reviewer 1 Report
The authors addressed all the previous concerns
no comments
Author Response
Response to Reviewer 1 Comments – Round 2
Point 1: Section 2.3 should be moved to the results.
Response 1: Section 2.3 was moved to the results.
Point 2: Decimal point should be used instead of decimal comma.
Response 2: Commas have been replaced by decimal points.
Point 3: Table 1, joint 1: it seems that the lesion increased in T6 compared to T0, is it correct?
Response 3: Not exactly, in joint 1 the tear at T0 was 10 mm, at 6 weeks (T6w) the blind examiner evaluated it with 10.5 mm, which is an acceptable difference in an ultrasound method, and indicative of no improvement. However, at 6 months (T6m), the lesion measured 0 mm, demonstrating complete healing. This difference of 0.5 mm was incorporated in the calculation of the size of the tears by time, as mentioned in the section “Size of the rupture in Groups A and B”.
Point 4: MRI is usually used to diagnose rotator cuff tendon tears and healing. This point should be specified/discussed and a hint must be added within the limits of the study.
Response 4: This point was added to the study limitations.
Another study limitation was the absent of the MRI method in evaluating PRCT. The use of ultrasound presents an effective alternative to the commonly employed magnetic resonance imaging (MRI). Despite MRI being renowned for its high resolution and detail, recent literature highlights the potential of ultrasound as an equally effective tool.
Moreover, in a meta-analysis, Liu et al., which included 144 diagnostic studies, revealed that high-field MRI had the highest diagnostic value in detecting any type of rotator cuff tendon lesion, followed by low-field MRI, high-frequency ultrasound, lowfield MRI, and low-frequency ultrasound. However, the differences in sensitivity and specificity between high-frequency ultrasound and MRI were quite small, which may be a reason to avoid the potential cost of MRI. Therefore, when making final decisions, the available equipment (MRI and ultrasound parameters) and the examiner's experience should also be taken into consideration [68].
A salient advantage of ultrasound is its accessibility and expedited operation, which proves to be beneficial in daily situation or in areas where access to MRI equipment is constrained. However, the selection of the optimal diagnostic tool should be primarily guided by the specific clinical scenario at hand.
While MRI maintains its position as the standard reference for assessing PRCT, ultrasound has emerged as a viable and beneficial alternative. More extensive research is required to establish clear guidelines on the most effective usage of ultrasound in the as-sessment of PRCT healing.
Reference:
- Liu F.; Dong J.; Shen W.J.; Kang Q.; Zhou D.; Xiong F. Detecting Rotator Cuff Tears: A Network Meta-analysis of 144 Diagnostic Studies. Orthop J Sports Med 2020, 5;8(2):2325967119900356.